# A gene-specific *T2A-GAL4* library for *Drosophila*

Pei-Tseng Lee[1], Jonathan Zirin[2], Oguz Kanca[1], Wen-Wen Lin[1], Karen L Schulze[1,3], David Li-Kroeger[1], Rong Tao[2], Colby Devereaux[2], Yanhui Hu[2], Verena Chung[2], Ying Fang[1], Yuchun He[1,3], Hongling Pan[1,3], Ming Ge[1], Zhongyuan Zuo[1,4], Benjamin E Housden[2], Stephanie E Mohr[2,5], Shinya Yamamoto[1,4,6,7], Robert W Levis[8], Allan C Spradling[8], Norbert Perrimon[2,5], Hugo J Bellen[1,3,4,6,7]*

[1]Department of Molecular and Human Genetics, Baylor College of Medicine, Houston, United States; [2]Department of Genetics, Harvard Medical School, Boston, United States; [3]Howard Hughes Medical Institute, Baylor College of Medicine, Houston, United States; [4]Jan and Dan Duncan Neurological Research Institute, Texas Children's Hospital, Houston, United States; [5]Howard Hughes Medical Institute, Harvard Medical School, Boston, United States; [6]Program in Developmental Biology, Baylor College of Medicine, Houston, United States; [7]Department of Neuroscience, Baylor College of Medicine, Houston, United States; [8]Department of Embryology, Howard Hughes Medical Institute, Carnegie Institution for Science, Baltimore, United States

*For correspondence: hbellen@bcm.edu

**Abstract** We generated a library of ~1000 *Drosophila* stocks in which we inserted a construct in the intron of genes allowing expression of *GAL4* under control of endogenous promoters while arresting transcription with a polyadenylation signal 3' of the GAL4. This allows numerous applications. First, ~90% of insertions in essential genes cause a severe loss-of-function phenotype, an effective way to mutagenize genes. Interestingly, 12/14 chromosomes engineered through CRISPR do not carry second-site lethal mutations. Second, 26/36 (70%) of lethal insertions tested are rescued with a single *UAS*-cDNA construct. Third, loss-of-function phenotypes associated with many *GAL4* insertions can be reverted by excision with *UAS-flippase*. Fourth, *GAL4* driven *UAS-GFP/RFP* reports tissue and cell-type specificity of gene expression with high sensitivity. We report the expression of hundreds of genes not previously reported. Finally, inserted cassettes can be replaced with *GFP* or any DNA. These stocks comprise a powerful resource for assessing gene function.

DOI: https://doi.org/10.7554/eLife.35574.001

## Introduction

Knowing where a gene is expressed and where the encoded protein is localized within the cell provides critical insight into the function of almost any gene (*Kanca et al., 2017*). The use of antibodies and molecular manipulation of genes have provided key tools to assess gene expression and protein localization in *Drosophila*. For example, thousands of *P*-element mediated enhancer detectors have been used to assess expression patterns (*Bellen et al., 2011*; *Bellen et al., 1989*; *Bier et al., 1989*; *O'Kane and Gehring, 1987*; *Wilson et al., 1989*). The original enhancer trap vectors were based on the presence of a relatively weak, neutral promoter driving *lacZ* that can be acted upon by adjacent enhancers as *P* elements often insert in 5' regulatory elements (*Bellen et al., 2011*; *Spradling et al., 2011*). In adapting a powerful binary expression system first developed in yeast (*Fischer et al., 1988*) for use in Drosophila, *Brand and Perrimon (1993)* replaced *lacZ* with *GAL4* to induce

**eLife digest** Determining what role newly discovered genes play in the body is an important part of genetics. This task requires a lot of extra information about each gene, such as the specific cells where the gene is active, or what happens when the gene is deleted. To answer these questions, researchers need tools and methods to manipulate genes within a living organism.

The fruit fly *Drosophila* is useful for such experiments because a toolbox of genetic techniques is already available. Gene editing in fruit flies allows small pieces of genetic information to be removed from or added to anywhere in the animal's DNA. Another tool, known as GAL4-UAS, is a two-part system used to study gene activity. The GAL4 component is a protein that switches on genes. GAL4 alone does very little in *Drosophila* cells because it only recognizes a DNA sequence called UAS. However, if a GAL4-producing cell is also engineered to contain a UAS-controlled gene, GAL4 will switch the gene on.

Lee et al. used gene editing to insert a small piece of DNA, containing the GAL4 sequence followed by a 'stop' signal, into many different fly genes. The insertion made the cells where each gene was normally active produce GAL4, but – thanks to the stop signal – rendered the rest of the original gene non-functional. This effectively deleted the proteins encoded by each gene, giving information about the biological processes they normally control.

Lee et al. went on to use their insertion approach to make a *Drosophila* genetic library. This is a collection of around 1,000 different strains of fly, each carrying the GAL4/stop combination in a single gene. The library allows any gene in the collection to be studied in detail simply by combining the GAL4 with different UAS-controlled genetic tools. For example, introducing a UAS-controlled marker would pinpoint where in the body the original gene was active. Alternatively, adding UAS-controlled human versions of the gene would create humanized flies, which are a valuable tool to study potential disease-causing genes in humans.

This *Drosophila* library is a resource that contributes new experimental tools to fly genetics. Insights gained from flies can also be applied to more complex animals like humans, especially since around 65% of genes are similar across humans and *Drosophila*. As such, Lee et al. hope that this resource will help other researchers shed new light on the role of many different genes in health and disease.

DOI: https://doi.org/10.7554/eLife.35574.002

expression of *UAS-effectors* (e.g. *GFP*, cDNAs, shRNAs). They showed that this technology allowed labeling of cells to assess gene expression patterns and drive expression of cDNAs (*Brand and Perrimon, 1993*). This binary system has been used to perform tissue-specific knockdown using *UAS*-RNAi constructs (*Dietzl et al., 2007*; *Ni et al., 2009*), carry out intersectional approaches to refine expression patterns in select neuronal populations via Split-GAL4 technology (*Luan et al., 2006*), perform stochastic neuronal labeling approaches via MARCM (Mosaic Analysis with a Repressible Cell Marker) (*Lee and Luo, 2001*), block synaptic transmission or induce neuronal excitation to assess neuronal activity (*Rosenzweig et al., 2005*; *Sweeney et al., 1995*), as well as numerous other manipulations (*Venken et al., 2011b*).

We previously developed the MiMIC (Minos-Mediated Insertion Cassette) technology to permit integration of any DNA cassette at a site where the MiMIC transposable element is inserted (*Venken et al., 2011a*). We created fly stocks with nearly 17,500 MiMIC insertions and characterized their properties (*Nagarkar-Jaiswal et al., 2015b*; *Venken et al., 2011a*). MiMICs contain two φC31 *attP* sites that can be used to exchange the integrated cassette with diverse cassettes containing two *attB* sites through Recombinase Mediated Cassette Exchange (RMCE) (*Bateman et al., 2006*; *Groth et al., 2004*; *Kanca et al., 2017*; *Nagarkar-Jaiswal et al., 2015b*; *Venken et al., 2011a*). We used RMCE to generate a library of protein trap lines where we inserted a cassette consisting of *SA (Splice Acceptor)-GFP-SD (Splice Donor)* (short for *SA-GSS-EGFP-FlAsH-StrepII-TEV-3XFlag-GSS-SD*, also abbreviated *GFSTF, GFP-tag*) into 400 MiMICs inserted in coding introns (introns flanked by two coding exons) (*Nagarkar-Jaiswal et al., 2015a*; *Nagarkar-Jaiswal et al., 2015b*). The synthetic *GFP* exon is spliced into the mRNA of the gene, leading to the translation of a protein with an internal *GFP* tag. This intronic *GFP* tagging approach allows us to determine which cells express the

corresponding gene/protein and assess subcellular protein distribution. Importantly, ~75% of intronically tagged genes appear functional (*Nagarkar-Jaiswal et al., 2015b*). These endogenous *GFP*-tagged lines provide an excellent tool to survey subcellular distribution of the encoded proteins. In addition, the *GFP* tagged proteins can be knocked down in a spatially and temporally restricted fashion, and loss of the *GFP*-tagged protein is reversible using the deGradFP technique as long as the gene is actively transcribed (*Caussinus et al., 2011*), allowing elegant in vivo manipulation (*Nagarkar-Jaiswal et al., 2015b*).

More recently, *Diao et al. (2015)* developed a *T2A-GAL4* technology, named *Trojan GAL4*, that integrates a cassette consisting of a *SA-T2A-GAL4-polyA* (polyadenylation signal) in coding introns of genes that carry MiMICs to assess the expression pattern of genes and measure or block neuronal activity (*Diao et al., 2015*; *Gnerer et al., 2015*). The *polyA* should arrest transcription of the gene in which the MiMIC is inserted, generating a truncated transcript. *T2A* is a viral ribosomal skipping site that arrests translation, which becomes reinitiated after the site, producing untagged GAL4 protein (*Diao and White, 2012*). The ability to replace intronic MiMICs with *T2A-GAL4* opens many avenues that are complementary to tagging genes that carry intronic MiMICs with *SA-GFP-SD* (the *GFSTF* tag). Indeed, *T2A-GAL4* could allow determination of expression patterns, notably including in tissues or cells where genes are expressed at such low levels that they cannot easily be detected using the *GFSTF* tag approach. Although, driving *UAS-GFP* with *GAL4* amplifies expression levels and greatly increases sensitivity, subcellular localization information is lost. In addition, *SA-T2A-GAL4-polyA* should cause a severe loss-of-function mutation (i.e. a truncated transcript due to the *polyA* signal) unless the *SA* allows exon skipping (*Rueter et al., 1999*) or the truncated protein is functional. Moreover, integration of a transgene carrying a *UAS*-cDNA for the gene that is mutated (GOI, gene of interest) should rescue phenotypes induced by insertion of a *SA-T2A-GAL4-polyA* cassette, allowing quick and efficient structure-function analyses (*Bellen and Yamamoto, 2015*). Finally, numerous other manipulations based on *GAL4/UAS* technology can be explored to assess function including those of species homologues, to query neuronal connectivity, impair activity, ablate cells, or assess gene or cellular functions, as well as various other applications (*Kanca et al., 2017*; *Venken et al., 2011b*). So far, about 50 genes have been reported to be tagged with a *Trojan-GAL4* cassette (*Chao et al., 2017*; *Conway et al., 2018*; *Diao et al., 2015*; *Diao et al., 2016*; *Hattori et al., 2017*; *Krüger et al., 2015*; *Lee et al., 2018*; *Li et al., 2017*; *Liu et al., 2017*; *Poe et al., 2017*; *Skeath et al., 2017*; *Toret et al., 2018*; *Wu et al., 2017*; *Yoon et al., 2017*). Hence, the power and generality of this technology remains to be explored. The potential usefulness of a large collection of *T2A-GAL4* insertion fly stocks led us to create a large library; assess the features, properties, and robustness of the *T2A-GAL4* method; and explore some of the potential applications of the technology.

Here, we report the conversion of 619 intronic MiMICs with *T2A-GAL4*. Given that there are only ~1860 genes containing MiMICs inserted between coding exons that can be used for tagging with *T2A-GAL4* (*Nagarkar-Jaiswal et al., 2015b*), we tested a number of vectors for CRISPR-mediated integration and eventually developed a vector and an efficient, gene-specific protocol for *T2A-GAL4* insertion that we named CRIMIC (CRISPR-Mediated Integration Cassette). Using this approach, we tagged 388 genes using CRIMIC. We characterized genetic features associated with these *T2A-GAL4* insertions, document numerous novel expression patterns, and provide compelling evidence that this library of ~1000 strains will permit a wide variety of elegant and highly valuable genetic, cell biological, and neurobiological applications.

## Results

### Comparison of *GFSTF* and *Trojan-GAL4* tagging of MiMIC-containing genes

As a part of the Gene Disruption Project, we created and sequenced the flanks of ~15,660 MiMIC insertions (*Nagarkar-Jaiswal et al., 2015b*; *Venken et al., 2011a*). Of these 2854 are intronic insertions that permit tagging of 1862 different genes. We classified 1399 insertions as 'Gold' as they are predicted to tag all transcripts annotated in FlyBase, 550 are 'Silver' and tag more than 50% of all gene transcripts, whereas 193 are 'Bronze' and tag less than 50% of the transcripts. As some genes are tagged with multiple MiMICs, the total is greater than 1862. We prioritized the tagging of 881

genes that have one or more human homolog (DIOPT Score $\geq$4 (*Hu et al., 2011*)) and are part of the 'Gold' collection (*Nagarkar-Jaiswal et al., 2015b*; *Yamamoto et al., 2014*). In addition, 139 'Gold' MiMICs in genes with low-confidence orthologs (DIOPT Score $\leq$3) or not conserved in humans were also selected, along with a number of 'Silver' and 'Bronze' insertions (see Flypush: http://flypush.imgen.bcm.tmc.edu/MIMIC/lines.php). We successfully tagged 611 genes with *GFSTF* (*Nagarkar-Jaiswal et al., 2015a*; *Nagarkar-Jaiswal et al., 2015b*; *Venken et al., 2011a*), and 211 in this work. We previously showed that conversion of MiMICs with *GFSTF* allows for efficient tagging of genes that carry intronic MiMICs and that 90% of intronically *GFP*-tagged proteins show robust GFP signals in third instar larval brains (*Nagarkar-Jaiswal et al., 2015b*). However, staining of adult brains revealed robust expression in only ~19% of the *GFP*-tagged genes tested (114/611, *Figure 1—figure supplement 1*). To achieve higher adult brain expression we prioritized genes based on the presence of human homologs and converted 619 MiMIC insertions to *SA-T2A-GAL4-polyA* (see Flypush: http://flypush.imgen.bcm.tmc.edu/pscreen/rmce).

We generated both *GFP* (*GFSTF*) and *T2A-GAL4* tagged lines by converting the same original MiMIC line through RMCE and compared the expression patterns for 104 genes, to assess if expression was consistently increased. *Figure 1A* shows expression in third instar larvae and adult brains of four proteins tagged with GFP. The expression and localization of the proteins encoded by *nAChRalpha1*, *dpr15*, *Pxn* and *Gprk2* are easily detectable in third instar larval brains and ventral nerve cords, yet exhibit weak or no detectable signals in adult brains. In contrast, the gene expression pattern visualized using *T2A-GAL4* converted MiMICs and assayed with *UAS-mCD8::GFP* (*Figure 1B and C*) exhibits robust GFP signals in third instar and adult brains. This method of integrating the *T2A-GAL4* is very efficient and is less time consuming than integrating *GFSTF* (*Nagarkar-Jaiswal et al., 2015b*), as RMCE-mediated conversion events can be easily detected by scoring insertion events crossed to *UAS-2xEGFP* and screening for expression in any tissue in embryos, larvae, or adults (*Diao et al., 2015*).

We previously showed that genes tagged with *GFSTF* faithfully reproduce the expression and subcellular distribution pattern of all tagged proteins tested (*Nagarkar-Jaiswal et al., 2015b*). We confirmed this observation as the similarities between *GFSTF* localization (*Figure 1—figure supplement 1*) and published antibody staining for Cactus (*Zhou et al., 2015*), Rgk1 (*Murakami et al., 2017*), Discs large 1, and Bruchpilot (*Nagarkar-Jaiswal et al., 2015b*) in the brain are obvious. However, *GAL4* strongly amplifies the expression of *UAS-mCD8::GFP* when compared to the endogenous GFP tagged proteins but the subcellular protein distribution is lost. As shown in *Figure 1—figure supplement 2*, in non-neuronal tissue the expression patterns as gauged with mCD8::GFP driven by *T2A-GAL4* or antibody staining overlap significantly for *arm* in larval wing disc, *Mhc* in larval muscle, and *osa* in larval eye-antenna imaginal discs (*Figure 1—figure supplement 2A*) when assessed at low resolution. However, for *trio*, which encodes a Rho guanyl-nucleotide exchange factor that regulates filamentous actin, the expression patterns do not overlap extensively, even at low resolution. Trio is known to play a role in the mushroom body (MB) neurons (*Awasaki et al., 2000*) as well as in motor neurons at neuromuscular junctions (NMJs) (*Ball et al., 2010*). However, the localization of the Trio protein (*Figure 1—figure supplement 2A* red, bottom row) in the larval central brain and ventral nerve cord (VNC) appears different from the *GAL4 >UAS-mCD8::GFP* pattern since GFP is strongly expressed throughout the MB and VNC, whereas the expression of Trio protein is low in VNC and the protein is localized to NMJs (red staining, insert). Similarly, we observe that mCD8::GFP driven by *T2A-GAL4* is also present at the NMJs (green staining, inset). In summary, the data are consistent and suggest that Trio is expressed in many neurons, including the motor neurons.

A comparison of the expression patterns of four genes tagged with both *GFSTF* and *T2A-GAL4>mCD8::GFP* exemplifies differences in the expression patterns. As shown in *Figure 1—figure supplement 2B*, the patterns of *SIFaR*, *zip*, *VGlut* and *mbl* are difficult to reconcile without further characterization. In summary, both the *T2A-GAL4* and the *GFSTF* conversions provide valuable information and should permit different applications.

## CRISPR-mediated insertion of MiMIC-like vectors

In order to vastly expand the collection of MiMIC-tagged genes, we initially tried to use CRISPR technology to insert MiMIC-like constructs and developed two vectors, pM14 and pM36. pM14 contains a MiMIC-like cassette (*attP-FRT-SA-3XSTOP-polyA-3xP3-EGFP-FRT-attP*) whereas pM36 lacks

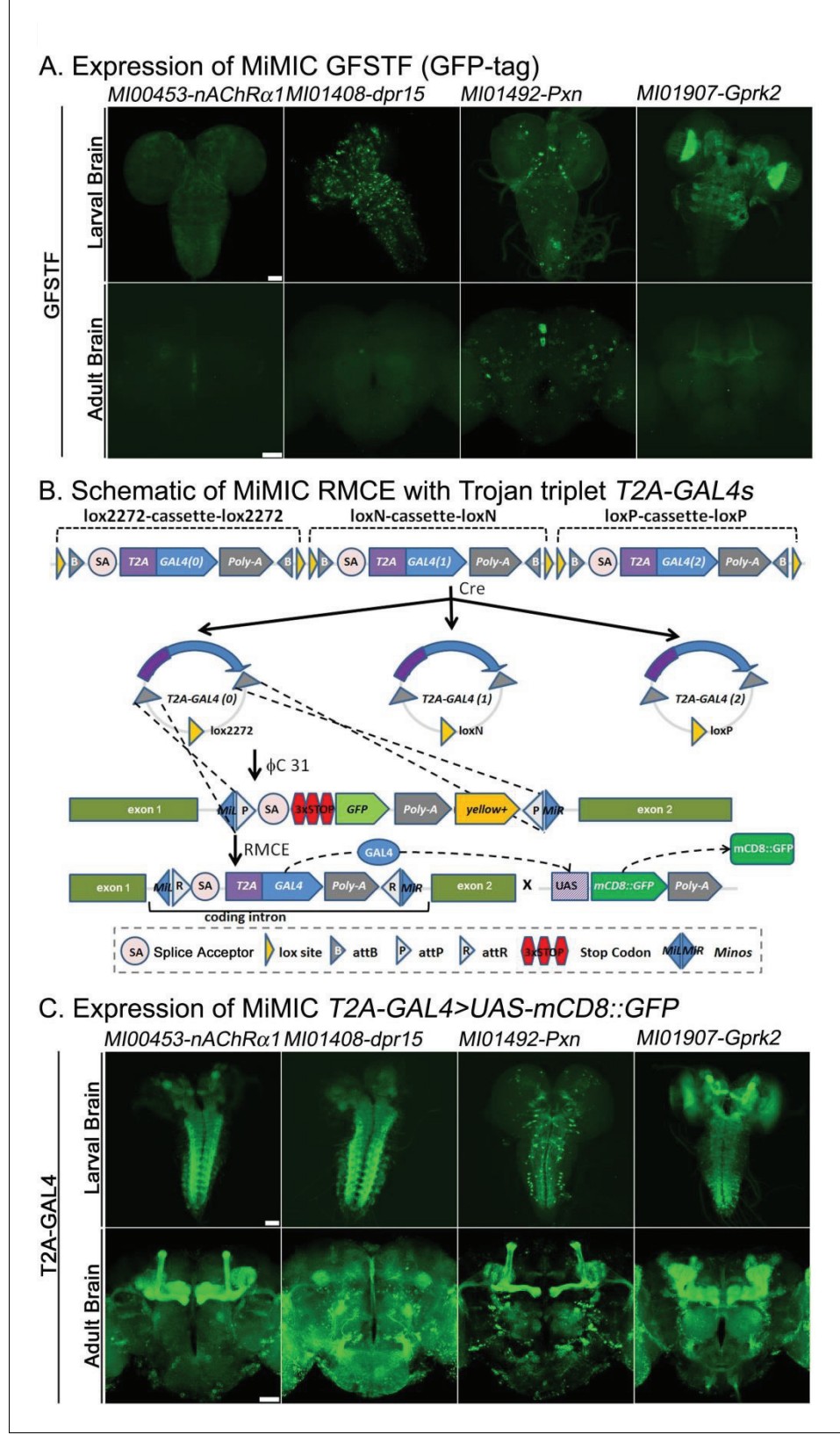

**Figure 1.** Protein distribution and expression patterns of genes containing MiMICs tagged with *GFSTF* or *T2A-GAL4*. The MiMIC transposon contains two inverted *attP* sites that allow RMCE. (**A**) Detection of the expression domains of the indicated genes tagged with *GFSTF* in larvae and adult brains. GFP: green (**B**) Schematic of the
*Figure 1 continued on next page*

*Figure 1 continued*

MiMIC conversion with Trojan triplet *T2A-GAL4* cassettes (*Diao et al., 2015*). Only the inserted *T2A-GAL4* cassette with the correct orientation and phase results in GAL4 expression that drives *UAS-mCD8::GFP* expression. (**C**) Detection of the expression domains in larvae and adult brains of genes tagged with *T2A-GAL4* using *UAS-mCD8::GFP*. mCD8::GFP: green. Scale bar: 50 μm.

DOI: https://doi.org/10.7554/eLife.35574.003

The following figure supplements are available for figure 1:

**Figure supplement 1.** Expression of *MiMIC GFSTFs* tagged genes in adult brains.

DOI: https://doi.org/10.7554/eLife.35574.004

**Figure supplement 2.** Similarities and differences between expression patterns associated with *GAL4 >UAS* GFP driven patterns and endogenous proteins in adult brains.

DOI: https://doi.org/10.7554/eLife.35574.005

the *FRT* sites present in pM14 (*Figure 2A*). Homology arms approximately 500–1000 bp in length were added to each side of these cassettes by Golden Gate Assembly (GGA) (*Engler et al., 2008*) to generate donor plasmids for homology directed repair (*Figure 2B*).

To ensure similar and clean genetic backgrounds for all transformation experiments, we isogenized the second and third chromosomes of the *nos-Cas9* flies into which we injected our constructs. We used the FindCRISPR tool which is based on a pre-computed database of CRISPR sgRNA designs requiring the presence of a PAM sequence at the end and a unique seed region (*Housden et al., 2015*). All sgRNA designs used the reference genome from FlyBase. Homology arms were amplified from genomic DNA from the isogenized *nos-Cas9* injection lines.

The mix of sgRNAs and donor vectors was injected into embryos expressing *Cas9*, under the *nanos* promoter (*nos-Cas9*), to ensure germline expression (*Kondo and Ueda, 2013*; *Ren et al., 2013*) for integration into introns of the GOI in a directional manner (*Casini et al., 2015*). We injected constructs for 89 genes with pM14 with a success rate of 57%, and 114 genes with pM36 with a success rate of only 26% (*Figure 2—figure supplement 1A*). The insertion efficiencies of these constructs were deemed too low, and thus they are no longer used in our production pipeline.

## CRISPR-mediated insertion of *T2A-GAL4* cassettes

The utility of the *T2A-GAL4* lines generated by RMCE of MiMICs encouraged us to use CRISPR/*Cas9* (*Zhang et al., 2014*) to insert *SA-T2A-GAL4-polyA* in introns of GOI using the CRISPR/*Cas9* system, similar to the T-GEM vector developed by *Diao et al. (2015)*. However, we added flanking FRT sites to allow excisions of the cassette with Flippase. We therefore designed a set of vectors with a swappable MiMIC-like cassette that contains *attP-FRT-SA-T2A-GAL4* (with phases 0, 1, and 2)*-polyA-3xP3-EGFP-FRT-attP* named pM37 (*Figure 3A*).

Upon many trials we settled on injecting 25 ng/μl of a single sgRNA and 150 ng/μl of the *-SA-T2A-GAL4-polyA-* donor construct (pM37) with ~1 kb homology arms on either side in isogenized *nos-Cas9* flies (*Housden et al., 2016*; *Housden and Perrimon, 2016*). As summarized in *Figure 4A*, we injected approximately 500 embryos for each of 557 different genes. The fly crosses for each target chromosome are documented in *Supplementary file 1*. The percentage of injected embryos surviving to first instar was 23% and on average 4.6 flies expressing GFP in the eye (*3xP3-GFP*) were recovered per injection. Molecular analysis of lines started from each individual GFP+ fly revealed that at least one insertion in the GOI was obtained for nearly 70% of the genes (*Figure 4A*). All insertions were confirmed by PCR (see Materials and Methods or Flypush for protocols and corresponding primers; *Figure 2—figure supplement 1B*). Note that the efficiency is higher if we omit the data for genes that map to the third chromosome as the *nos-Cas9* transgene insertion on the second chromosome carries a recessive lethal mutation, reducing the efficiency significantly. Alternative *nos-Cas9* insertions on the second and X chromosomes are being tested to improve the efficiency.

To assess expression patterns of the GOIs, we crossed the transgenic flies to *UAS-mCD8::RFP*, which labels cell membranes (*Belenkaya et al., 2008*) and thus can be easily distinguished from the *3XP3-GFP* tag, which is used as a selectable marker for transgenesis and is sparsely expressed in the nervous system (*Figure 3B*). As shown in *Figure 3C*, the insertions in different genes produce a variety of expression patterns. For ten genes picked at random, several different independently isolated

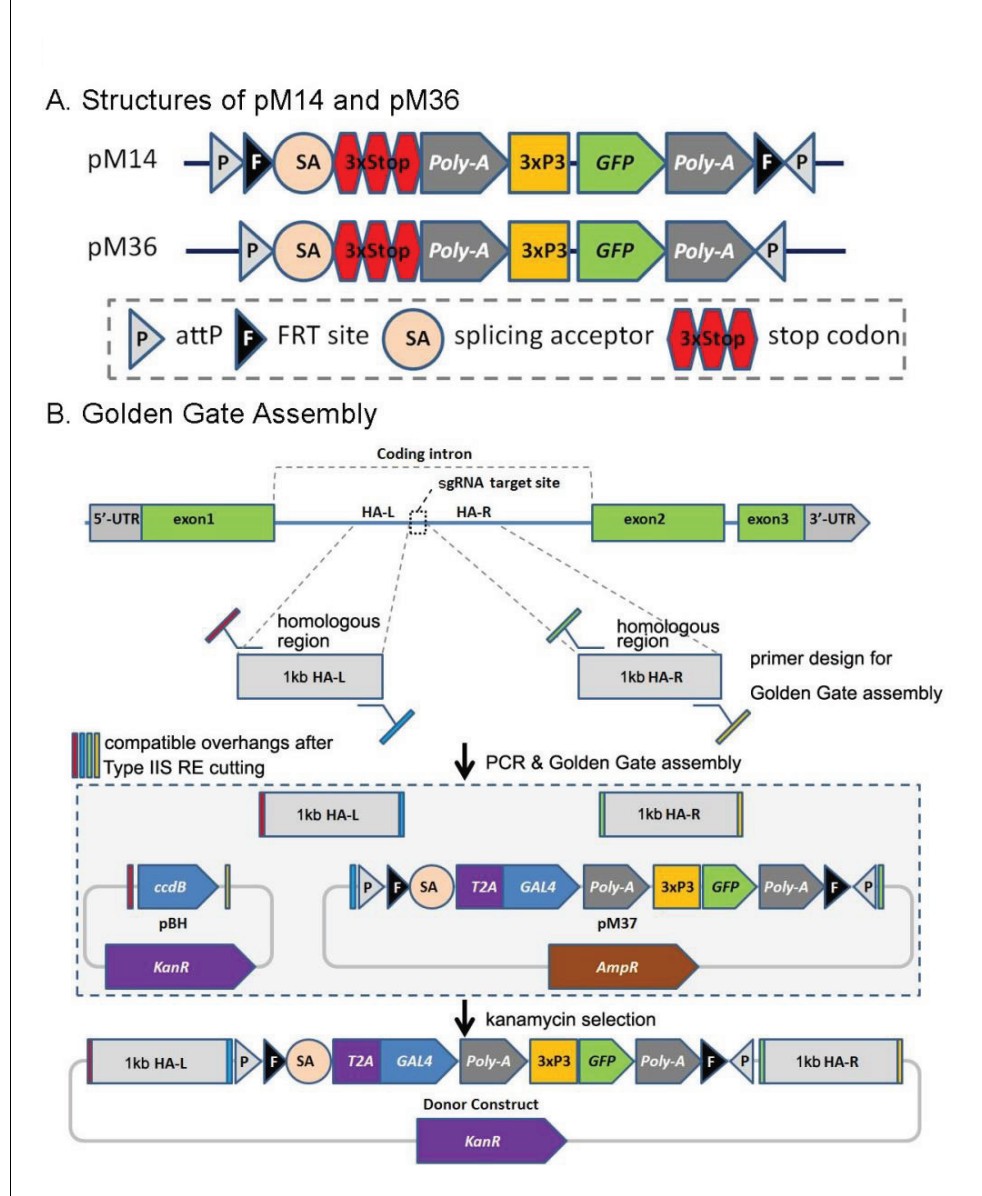

**Figure 2.** CRIMIC pM14 and pM36, and Golden Gate Assembly. (**A**) Structures of pM14 and pM36. The CRIMIC pM14 cassette contains MiMIC-like cassette (*SA-3xstop-polyA*) and two *FRT* sites. The CRIMIC pM36 cassette was modified by removing the two *FRT* sites from PM14. (**B**) Golden Gate Assembly. Two sets of primers containing Type IIS RE sites are typically used to amplify ~1 kb homology arms by PCR. These arms, pM37 DNA and pBH vector (KanR) digested with Type IIS Restriction Enzymes and cloned using Golden Gate Assembly to generate the donor construct in a single reaction. The pM14/pM36 based donor DNAs were constructed with the same approach. The complete donor construct is selected with kanamycin. The components in these diagrams are not drawn to scale.

DOI: https://doi.org/10.7554/eLife.35574.006

The following figure supplement is available for figure 2:

**Figure supplement 1.** The efficiency of cassette insertion with CRIMIC pM14 and pM36, and PCR validation.
DOI: https://doi.org/10.7554/eLife.35574.007

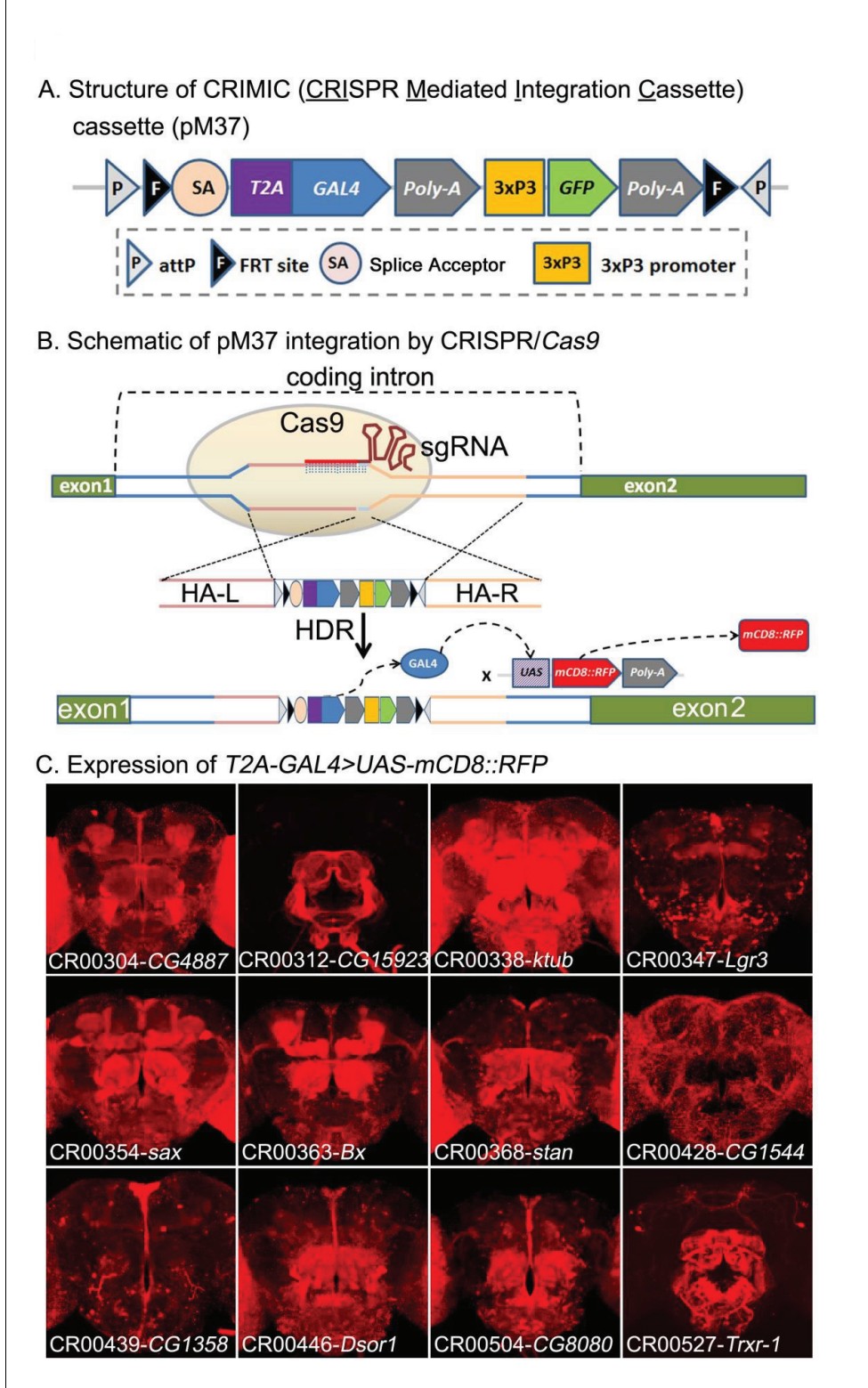

**Figure 3.** CRIMIC: *T2A-GAL4* integration using CRISPR and expression patterns of tagged genes. (**A**) Structure of the CRIMIC pM37 cassette. (**B**) Schematic of the CRIMIC insertion strategy through two 1 kb homology arms by HDR (homology directed repair) based on CRISPR/*Cas9* technology. (**C**) Expression patterns observed in adult fly brains of *T2A-GAL4 > UAS-mCD8::RFP*. mCD8::RFP (red). Scale bar: 50 μm.

DOI: https://doi.org/10.7554/eLife.35574.008

and sequenced insertions for a given gene exhibited very similar expression patterns, suggesting that the method is robust.

## Coding intronic insertions of the *SA-T2A-GAL4-polyA* cassette generate loss-of-function mutations for ~90% of insertions

The design of pM37 and the ability to use CRISPR should provide the following advantages: first, the ability to insert the CRIMIC cassettes in sites that affect all transcripts encoded by a gene and create severe loss-of-function or null alleles (*Figure 4—figure supplement 1A*); second, the ability to excise the mutagenic cassette in vivo (revert) using *UAS-FLP* under the control of *GAL4* inserted in the GOI to assess if the CRIMIC cassette is indeed responsible for the observed phenotypes (*Figure 4—figure supplement 1B*); third, the ability to revert loss-of-function phenotypes in any tissues at any time to assess when a protein is required and if loss of the gene causes a permanent or reversible phenotype at the time of excision; fourth, the ability to choose an integration site that does not disrupt protein domains upon retagging with *GFSTF* (*Figure 4—figure supplement 1C*); fifth, the ability to insert any DNA flanked by *attB* sites and replace the *SA-T2A-GAL4-polyA* cassette. These include the following available cassettes: *GFSTF*, *mCherry*, *GAL80*, *LexA*, *QF*, and *split-GAL4* (*Diao et al., 2015*; *Venken et al., 2011a*). Finally, the ability to test for rescue of the mutant phenotypes by driving the corresponding *UAS*-cDNA, a feature that also allows for structure-function analysis (*Figure 4—figure supplement 1D*).

Insertion of a *SA-T2A-GAL4-polyA* in a coding intron should arrest transcription at the *polyA* signal (PAS or AATAAA) unless the site is masked (*Berg et al., 2012*). Hence, MiMIC and CRIMIC *T2A-GAL4* insertions should cause a severe loss-of-function mutation in most but not all cases, depending on where the *SA-T2A-GAL4-polyA* is inserted and whether or not all transcripts are effectively disrupted by the cassette (*Figure 4—figure supplement 1A*). To test the mutagenic capacity of the *T2A-GAL4* cassette, we selected insertions in 100 genes (82 MiMIC-derived insertions and 18 CRIMICs, *Supplementary file 2*) that are annotated in FlyBase (http://flybase.org/) as essential genes, based on previous publications. Of these, 80 were categorized as 'Gold', 14 as 'Silver' and six as 'Bronze' (*Supplementary file 2*). We performed complementation tests using 99 molecularly defined deficiencies (Dfs) that remove the affected gene (*Parks et al., 2004*; *Ryder et al., 2004*) and one *P*-element insertion for *Cka* (*Supplementary file 2*). As shown in *Figures 4B*, 90 insertions fail to complement the lethality, five are semi-lethal (less than 5% escapers), and five are viable (see Discussion).

Because the *SA-T2A-GAL4-polyA* cassette should prematurely terminate transcription, and as the cassette in CRIMICs is flanked by *FRT* sequences, we next tested if the lethality associated with eleven insertions can be reverted by using the GAL4 to drive *UAS-FLP* (*Figure 4—figure supplement 1B*). We tested excision of 11 CRIMIC *T2A-GAL4* insertions in essential genes on the X chromosome by simply crossing them with *UAS-FLP*. As shown in *Figure 4C*, eight out of eleven hemizygous lethal insertions on the X chromosome produced numerous viable flies when crossed to *UAS-FLP*. To assess the efficiency of FLP/FRT mediated CRIMIC cassette excision for the three genes for which we did not observe viable flies (*Dsor1*, *Raf* and *Marf*), we tested if the *T2A-GAL4/+;+/+; UAS-FLP/+* females lacked the *3xP3-GFP* marker associated with the *T2A-GAL4* insertions. As shown in *Figure 4—figure supplement 2*, these flies did not express or barely expressed GFP in the eye, indicating that the efficiency of FLP-mediated excision is high. Given the rescue failure, we also tested whether these lines carry second-site recessive lethal mutations. However, all three are rescued by a genomic *P[acman]* clone (*Table 1*) indicating that these chromosomes do not carry second-site lethal mutations. All together, we conclude that cassette excision can revert the phenotype in most cases, providing a simple and powerful tool to assess the requirement for a gene product in a variety of cells and assess if the phenotype of interest is caused by the loss-of-function of the GOI (see Discussion).

## Expression of *UAS*-cDNA rescues lethality associated with *SA-T2A-GAL4-polyA* insertions for ~70% of genes

Expression of *GAL4* may allow rescue of the lethality associated with an insertion by driving expression of a *UAS-cDNA* in a pattern that corresponds to the gene (*Figure 4—figure supplement 1D*). However, this may not be effective in many cases as the vast majority of genes have more than one

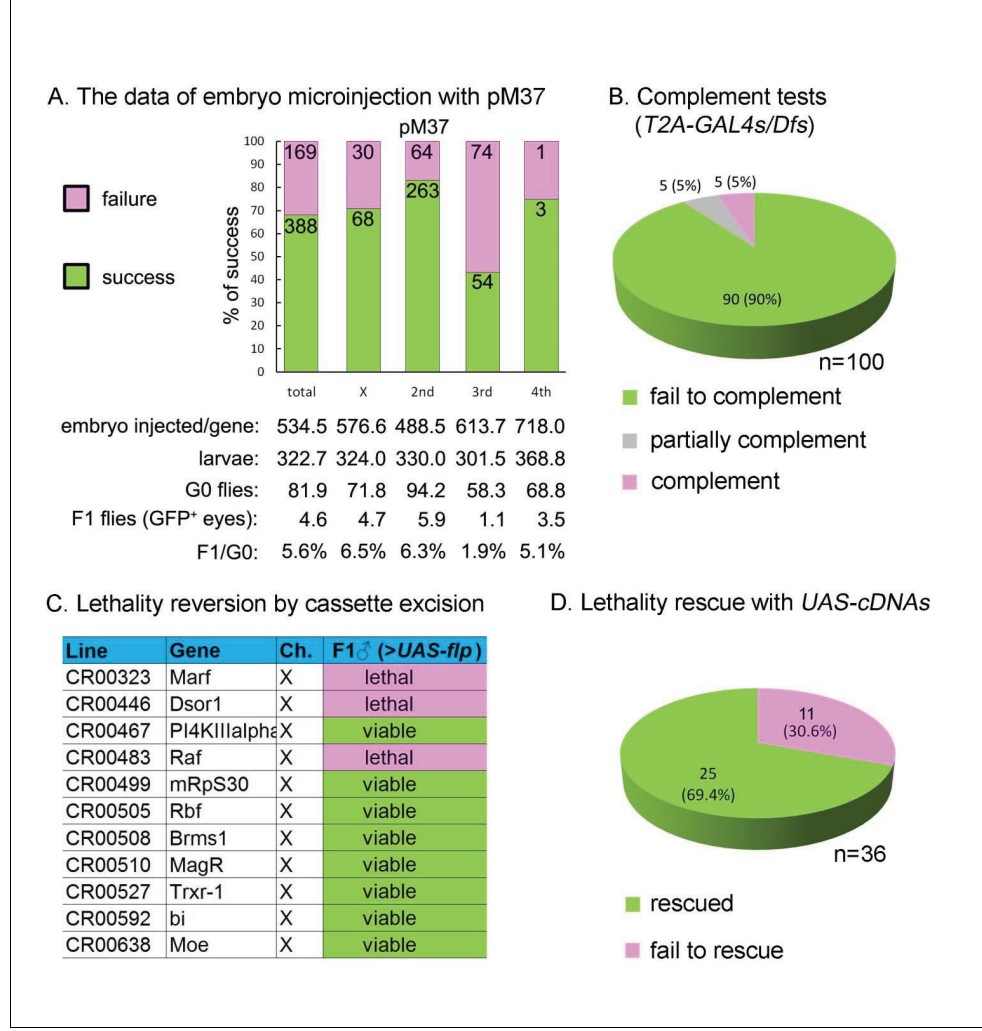

**Figure 4.** Summary of CRIMIC *T2-GAL4* integration efficiency and genetic properties of *T2A-GAL4* insertions (**A**) microinjection success rates for pM37. (**B**) Complementation test: 90% of the *T2A-GAL4* containing chromosomes fail to complement the corresponding Dfs; 5% produced less than 1/3 of the expected progeny; and 5% fully complemented the Dfs. For details see Supplemental Information 2. (**C**) *T2A-GAL4* cassette excision. The lethality associated with 8 out 11 insertions is reverted in the presence of *UAS-FLP*. (**D**) Rescue of the lethality of the *T2A-GAL4* cassette insertions with *UAS*-cDNA.

DOI: https://doi.org/10.7554/eLife.35574.009

The following figure supplements are available for figure 4:

**Figure supplement 1.** Applications of the CRIMIC technology.
DOI: https://doi.org/10.7554/eLife.35574.010
**Figure supplement 2.** *T2A-GAL4* cassette excision upon *FRT-FLP*.
DOI: https://doi.org/10.7554/eLife.35574.011

splice isoform, and rescue with any one isoform encoded by a *UAS-cDNA* construct might not work (*Table 1*). In addition, many cDNAs are tagged at the C-terminal end and it has been estimated that about 22% of the genes tagged with 3XHA (*Bischof et al., 2013*) and 33% tagged with GFP disrupt gene function (*Sarov et al., 2016*). Moreover, since the *GAL4/UAS* system is an over-expression system, cDNA rescue may not be possible for genes that are sensitive to dosage. Nevertheless, we assessed the ability of a single *UAS-cDNA* per gene to rescue mutant phenotypes associated with disruption of 36 genes for which we were able to find a *UAS-cDNA* (*Bischof et al., 2013*; *Gramates et al., 2017*). For 11 genes on the X-chromosome, we assessed rescue of male lethality, whereas for genes on the second and third chromosomes, we assessed rescue of *SA-T2A-GAL4-*

*polyA*-induced lethality over the corresponding Dfs that fail to complement the lethality. To ensure that the lethality of the genes on the X-chromosome is indeed associated with the insertions, we first performed genomic rescue using the 80 kb *P[acman]* BAC transgenic lines (*Venken et al., 2010*). The lethality of all genes on the X-chromosome was rescued with the corresponding *P[acman]* clones (*Table 1*), indicating that these chromosomes are very unlikely to carry second-site mutations. Of the 36 essential genes that carry *SA-T2A-GAL4-polyA*, 25 (~70%) could be rescued by a single *UAS-cDNA* driven by the endogenous *GAL4* (*Figure 4D*; *Table 1*).

## Characterization of cell type-specific expression patterns of genes tagged with *T2A-GAL4*

The sensitivity of *T2A-GAL4* tagging allows us to determine where genes are expressed, especially when expression levels in specific cell populations are low, as shown for the adult brain in *Figure 1*. We therefore determined the expression patterns of 550 genes in adult brains and documented expression patterns of many genes that have not been previously reported (*Gramates et al., 2017*) (*Figure 5*; *Figure 5—figure supplements 1–3*). Nearly 80% of all tested tagged genes are expressed in adult brains.

The smallest category of genes (9/550 or 2%) corresponds to genes expressed in trachea, a tubular system that provides oxygen to all tissues (*Varner and Nelson, 2014*). For example, *breathless* (*btl*) encodes a protein kinase expressed specifically in the trachea and is involved in tracheal branching (*Lee et al., 1996*). A comparison of the *GAL4>UAS-mCD8::GFP* expression pattern of a GAL4 based *P*-element enhancer detector in *btl* (*P[GawB]btl^NP6593*) (*Hayashi et al., 2002*) and the *T2A-GAL4* insertion (*MI03286-TG4.0*) in the brain and thoracicoabdominal ganglion (TAG) show very similar mesh-like tracheal patterns. Another gene previously documented to be expressed in trachea, *empty spiracles* (*emp*), also shows that the *T2A-GAL4* insertion drives expression in trachea (*Hart and Wilcox, 1993*). In *Figure 5* and *Figure 5—figure supplement 1*, we report the expression of seven other genes that have not been reported to be expressed in trachea (FlyBase 2.0/FB2017_06). Hence, nine genes out of 550 tested are expressed in trachea and for seven of these, detection of expression in the trachea is novel (*Frl, CG8213, sprt, geko, ex, Samuel, Cad96Ca*).

The next most frequent category consists of genes whose expression are mostly confined to a subtype of cells corresponding to glia. Glia account for about 10% of the cells in the fly brain (*Kremer et al., 2017*) and about 50% of cells in the mammalian brain (*von Bartheld et al., 2016*). To assess various glial patterns in the brain upon *UAS-mCD8::GFP* expression, we selected five known glial cell *GAL4* drivers as controls: *repo-GAL4* (all glia except midline glia), *gcm-GAL4* (embryonic glia), *NP2222-GAL4* (cortex glia), *NP6520-GAL4* (ensheathing glia) and *NP1243-GAL4* (astrocyte-like glia) (*Awasaki and Lee, 2011*). We identified 19/550 genes that are mostly or specifically expressed in one or several types of glia cells. Seven were previously shown to be expressed in glia: *CIC-a, loco, CG10702, CG6126, Gs2, Egfr* and *Tret1-1* (*Figure 5*; *Figure 5—figure supplement 2*), whereas 12 have not previously been associated with glial expression based on available data (*bdl, Zasp52, rols, ine, CG5404, CG14688, CG31663, ry, CG4752, βTub97EF, CG32473, LManII*; *Figure 5* and *Figure 5—figure supplement 2*). Note that *ry (rosy)* is known to be expressed in pigment cells of the eye (*Keller and Glassman, 1965*), and that these cells function as glial cells in this organ (*Liu et al., 2017*).

Finally, about 80% of lines showed expression patterns in adult brain neurons. Given the complexity of the brain and the sheer number of different expression patterns in neurons, we decided to focus on a single neuronal population that is easily identifiable and on expression patterns that were not previously documented. We selected the neurons of the pars intercerebralis (PI), which are located on the dorsal medial side of the brain and project to the tritocerebrum and the corpora cardiaca in the middle central area (*Nässel et al., 2013*). They secrete a variety of neuropeptides as well as *Drosophila* Insulin Like Peptides or DILPs (*Rulifson et al., 2002*). This cluster of neurons is a neuroendocrine command center that not only controls cell growth by releasing DILPs but also controls fly behaviors, including aggression, via secreted neuropeptides (*Davis et al., 2014*; *de Velasco et al., 2007*). The gene *Ilp2* encodes Insulin-like peptide 2. An *Ilp2* promotor *GAL4* fusion (*P{Ilp2-Gal4})* (*Broughton et al., 2005*) was used to express mCD8::GFP in a subset of PI neurons as a positive control. The expression of GFP in PI neurons driven by *T2A-GAL4* insertions in *AstA-R2* (*Allatostatin A receptor 2*) and *Lkr* (*Leucokinin receptor*), agrees well with previous observations of their expression in these neurons (*Cannell et al., 2016*; *Hentze et al., 2015*), In addition, we found 18

**Table 1.** Rescue of the lethality of *T2A-GAL4s* insertions/Dfs with *aUAS-cDNA* and genomic duplications with *P[acman]* clones. *1:(**Luo et al., 2017**)*2:(**Chao et al., 2017**)*3:(**Yoon et al., 2017**)*4:(**Sandoval et al., 2014**). Note that a failure to rescue lethality does not mean that it cannot partially rescue other scorable phenotypes.

| Line | Gene | Chr. | Protein isoforms | Flies for complementation test | Flies for rescue | |
|---|---|---|---|---|---|---|
| | | | | | Fly cDNA | Genomic DNA |
| MI01374-TG4.0 | sbr | X | 1 | NA | no tag | Dp(1;3)DC508 |
| MI02836-TG4.0 | cac*1 | X | 8 | NA | EGFP | Dp(1;3)DC131 |
| MI07818-TG4.0 | acj6 | X | 13 | NA | 3xHA | Dp(1;3)DC192 |
| MI08675-TG4.1 | arm | X | 2 | NA | 3xHA | Dp(1;3)DC034 |
| MI10323-TG4.1 | flw | X | 2 | NA | 1xHA | Dp(1;3)DC224 |
| MI12214-TG4.2 | if | X | 2 | NA | no tag | Dp(1;3)DC319 |
| MI00783-TG4.0 | stj | 2 | 3 | Df(2R)Exel7128/CyO | 3xHA | NA |
| MI02963-TG4.0 | CAP | 2 | 20 | Df(2R)BSC281/CyO | no tag | NA |
| MI03306-TG4.1 | kuz | 2 | 4 | Df(2L)BSC147/CyO | no tag | NA |
| MI03597-TG4.1 | mol | 2 | 2 | Df(2R)Exel6066/CyO | 3xHA | NA |
| MI04800-TG4.1 | lola | 2 | 20 | Df(2R)ED2076/SM6a | 3xHA | NA |
| MI06876-TG4.1 | spin | 2 | 3 | Df(2R)Jp8, w[+]/CyO | myc-EGFP | NA |
| MI09180-TG4.1 | Bsg | 2 | 2 | Df(2L)ED548/SM6a | 3xHA | NA |
| MI09585-TG4.1 | Lpt | 2 | 2 | Df(2R)BSC610/SM6a | 1xHA | NA |
| MI13162-TG4.0 | Rho1 | 2 | 1 | Df(2R)ED2457/SM6a | 3xHA | NA |
| MI13708-TG4.0 | Cka | 2 | 4 | P{ry[+t7.2]=PZ}Cka[05836] cn[1]/CyO | EGFP | NA |
| MI15480-TG4.2 | kn*2 | 2 | 5 | Df(2R)BSC429/CyO | 3xHA | NA |
| MI02220-TG4.1 | dally | 3 | 1 | Df(3L)ED4413/TM6C, cu[1] Sb[1] | no tag | NA |
| MI04910-TG4.1 | ftz-f1 | 3 | 3 | Df(3L)BSC844/TM6C, Sb[1] cu[1] | 3xHA | NA |
| MI06026-TG4.1 | Nc73EF*3 | 3 | 3 | Df(3L)ED4685/TM6C,cu[1] Sb[1] | Flag | Dp(1;3)DC245 |
| MI07056-TG4.0 | Atg1 | 3 | 2 | Df(3L)BSC613/TM6C, cu[1] Sb[1] | no tag | NA |
| MI08143-TG4.0 | Sod1 | 3 | 2 | Df(3L)BSC817/TM6C, Sb[1] cu[1] | no tag | NA |
| MI05068-TG4.0 | kdn | X | 2 | NA | NA | Dp(1;3)DC154 |
| Line | Gene | Chr. | Transcripts | Df | Fly cDNA | Genomic DNA |
| CR00323 | Marf | X | 2 | NA | 1xHA*4 | Dp(1;3)DC155 |
| CR00446 | Dsor1 | X | 2 | NA | 3xHA | Dp(1;3)DC205 |
| CR00483 | Raf | X | 1 | NA | no tag | Dp(1;3)DC404 |
| CR00505 | Rbf | X | 1 | NA | 3xHA | Dp(1;3)DC012 |
| CR00638 | Moe | X | 7 | NA | myc | Dp(1;3)DC199 |
| CR00354 | sax | 2 | 3 | Df(2R)BSC265/CyO | 3xHA | NA |
| CR00465 | Dap160 | 2 | 6 | Df(2L)BSC302/CyO | no tag | NA |
| CR00466 | Eps-15 | 2 | 4 | Df(2R)BSC606/SM6a | no tag | NA |
| CR00494 | l(2)gd1 | 2 | 2 | Df(2L)Exel6027/CyO | 1xHA | NA |
| CR00521 | Npc1a | 2 | 2 | Df(2L)BSC143/CyO | YFP | NA |
| CR00559 | Sod2 | 2 | 1 | Df(2R)Exel7145/CyO | no tag | NA |
| CR00587 | Hr38 | 2 | 2 | Df(3R)BSC510/TM6C, Sb[1] cu[1] | 3xHA | NA |
| CR00762 | Wee1 | 2 | 1 | Df(2L)BSC108/CyO | no tag | NA |
| CR00452 | sr | 3 | 4 | Df(3R)BSC510/TM6C, Sb[1] cu[1] | no tag | NA |

Blue: fail to complement

Gray: partially complement

Green: rescued

Pink: fail to rescue

Orange: rescue phenotype but not lethality

DOI: https://doi.org/10.7554/eLife.35574.012

genes (*CG31547*, *if*, *NimB2*, *Lerp*, *CG7744*, *cnc*, *CG2656*, *spin*, *gem*, *Fs*, *Aldh-III*, *CG33056*, *grsm*, *CG31075*, *Pi3K68D*, *Dh44-R2*, *Lgr4*, *Atg16*) that are expressed in PI neurons and yet have not been previously described as such (FlyBase 2.0/FB2017_06) (*Figure 5*; *Figure 5—figure supplement 3*).

## Discussion

Here, we report the creation of ~1000 *T2A-GAL4* lines by two different methods: 619 generated by RMCE of MiMIC insertions and 388 by CRIMIC, a novel CRISPR-mediated strategy. Our success rate of *MiMIC T2A-GAL4* conversion was 68.1% (543/797) upon a single attempt and 41.1% (76/185) upon a second attempt. Hence, we failed twice for 109 out of 797 genes. The *T2A-GAL4* insertions not only provide a *GAL4* driver that reveals the cells in which the targeted genes are expressed with great sensitivity but also allow many useful applications for testing gene function. We show that the CRIMIC technology is as powerful and reproducible as converting MiMICs with *T2A-GAL4,* and we should therefore be able to tag at least half of the genes in the *Drosophila* genome with the *T2A-GAL4* CRIMIC approach as they carry suitable introns that are large enough.

While the conversion of MiMICs depends on the presence of intronic MiMIC insertions, the CRIMIC approach allows us to select many genes that do not carry a MiMIC but contain an intron that is large enough and has proper sgRNA target sites to introduce a cassette that carries *SA-T2A-GAL4-polyA* flanked by *FRT* sites. The cloning success rate for the donor vector was about 80% on a first attempt, but significantly higher when repeated for another intron. This should allow us to tag about ~45–50% of all fly genes as those with short coding introns or without introns cannot be targeted using this strategy. By injecting ~535 embryos/construct we average a 70% successful integration rate. If we exclude the data for the third chromosome, where the *nos-Cas9* isogenized strain used was sub-optimal, our success rate is ~80%. We do not anticipate that we will be able to improve this much in the future except for the third chromosome. However, we are currently developing strategies with much shorter homology arms to avoid cloning and reduce the number of injected embryos, as our approach is labor-and cost-intensive. Indeed, we estimate that each line requires approximately ~50 hr of work for technicians, postdoctoral fellows, and bioinformaticians to obtain a single characterized stock deposited in the BDSC.

This technology is based on the properties of the *SA-T2A-GAL4-polyA* cassette. Issues with efficiency of those properties may limit the use of this cassette. First, skipping of the *SA* would reduce or abolish the gene-trap function of this cassette, leading to hypomorphic or neutral alleles of the GOI. The *SA* used here corresponds to intron 18 of *Mhc* (*Hodges and Bernstein, 1992*), a *SA* that has been used before (*Diao et al., 2015*; *Morin et al., 2001*; *Nagarkar-Jaiswal et al., 2015a*; *Nagarkar-Jaiswal et al., 2015b*; *Venken et al., 2011a*; *Zhang et al., 2014*). We show that this *SA* is quite effective, as lethal insertions in essential genes fail to complement the lethality of known alleles and deficiencies in 90% of the cases tested. These data also indicate that a second feature of the cassette, the *polyA* signal, is efficient at arresting transcription. As previously shown for a few genes (*Diao et al., 2015*; *Gnerer et al., 2015*), *GAL4* drives *UAS-GFP* or *RFP* expression efficiently in all cases tested and permits detection of expression in cells that express low mRNA and protein levels (*Figure 1* and *Figure 3*). Although the *GAL4/UAS* binary system strongly enhances the detection sensitivity when compared to the expression of the endogenous gene in the adult head tagged with *GFSTF*, this is much less the case in the third instar larval CNS (*Figure 1* and *Figure 1—figure supplement 2B*) (*Nagarkar-Jaiswal et al., 2015b*). We have no obvious explanation for this discrepancy. In summary, although it is impossible to prove that the *GAL4* is faithfully mimicking the endogenous expression given its enhanced sensitivity, the data we have compiled so far indicate that these insertions accurately represent the expression of the vast majority of genes.

The latter feature is important, as current *GAL4*-driver resources developed at the Janelia Research Campus and Vienna Drosophila Resource Center (*Jenett et al., 2012*; *Pfeiffer et al., 2008*) are based on very different premises. The driver transgenes were engineered to label few neurons. Indeed sparse labeling is a prerequisite to study neural networks. Given that the regulatory elements of genes used to create these collections are removed from their endogenous context it is difficult to determine which enhancers mimic a portion of the expression pattern of the gene they

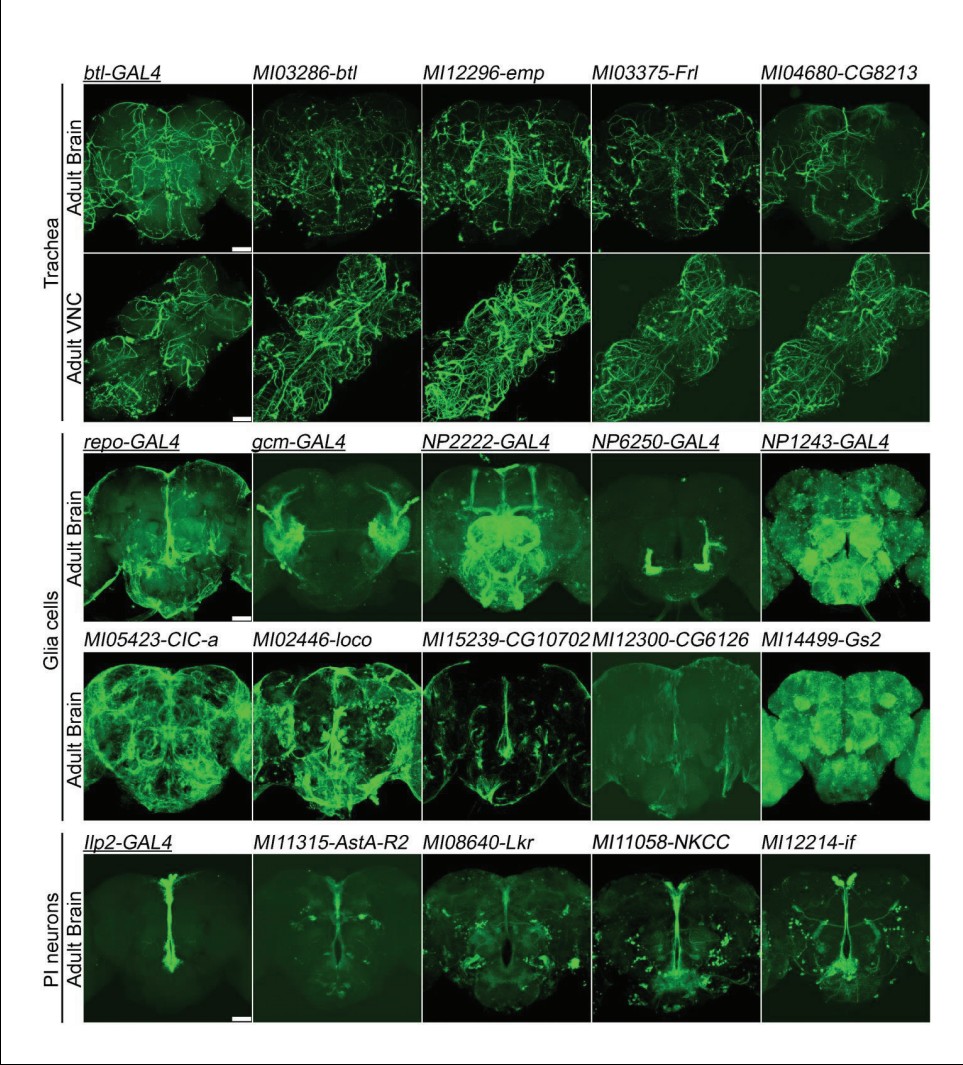

**Figure 5.** Genes expressed in (**A**) trachea, (**B**) glial cells, and (**C**) Pars Intercerebralis Neurons based on *T2-GAL4* insertions. The *GAL4s* (underlined) are existing *P*-element enhancer traps expressing *GAL4* in specific cell populations and serve as controls. mCD8::GFP: green. Scale bar: 50 μm.

DOI: https://doi.org/10.7554/eLife.35574.013

The following figure supplements are available for figure 5:

**Figure supplement 1.** Genes specifically expressed in trachea.

DOI: https://doi.org/10.7554/eLife.35574.014

**Figure supplement 2.** Genes expressed in glia.

DOI: https://doi.org/10.7554/eLife.35574.015

**Figure supplement 3.** Genes expressed in PI neurons.

DOI: https://doi.org/10.7554/eLife.35574.016

have been derived from, as repressors may not be present and enhancers may be truncated or not tested (FlyLight) (*Jory et al., 2012*). Hence, it should now be possible to compare the patterns of the genes presented here with those based on *GAL4* patterns driven by the ~2–3 kb fragments used in these studies (*Jory et al., 2012*; *Pfeiffer et al., 2008*).

Given the caveats associated with CRISPR technology (*Doench et al., 2016*), it is important to demonstrate that an observed phenotype is indeed associated with the insertion. In addition, we have previously shown that the genetic manipulations based on MiMIC can induce a significant number of second-site mutations (*Nagarkar-Jaiswal et al., 2015b*; *Venken et al., 2011a*). We therefore

attempted to rescue the lethal phenotypes associated with CRIMIC *T2A-GAL4* insertions with *UAS-FLP,* as this should excise the cassette. We found that 8 of 11 CRIMIC insertions that cause lethality were reverted with *UAS-FLP* (*Figure 4C*), providing a quick tool to assess genetic background load. The results of this experiment also indicate that the cassette can be excised with other *FLP* drivers like *LexA* or *hsp70* promoter driven *FLP*. Hence, most chromosomes engineered through CRISPR in this study do not carry second-site lethal mutations and this was confirmed with genomic *P[acman]* rescue constructs: all mutations tested were rescued with the corresponding *P[acman]* clones (*Venken et al., 2010*) (*Table 1*). The data also indicate that the delay between FLP production by GAL4 and excision is not critical for most essential genes. Finally, we note that the failure to rescue lethality was not due to a failure of excision for *Dsor1*, *Raf* and *Marf*. Indeed, flies that express GAL4 and FLP lack GFP expression in the eyes (*Dsor1*) or produce very little GFP derived from the *3xP3-EGFP* marker (*Raf* and *Marf*) (*Figure 4—figure supplement 2*), suggesting that excision of the *T2A-GAL4* cassette was successful in all cases tested. Hence, tissue-specific excision should easily be induced using *hs-FLP* or another binary system (*Venken et al., 2011b*), allowing one to perform conditional rescue experiments and assess in some cases when and where proteins are required. In summary, combining the features of *T2A-GAL4* with the FLP-mediated excision system provides numerous possibilities.

One of the most useful applications of *T2A-GAL4* may be the ability to use *SA-T2A-GAL4-polyA* with *UAS-cDNAs* to perform structure-function analyses, that is, test the consequences of removal of protein domains and/or of introducing point mutations into the *UAS-cDNA* construct, or even to test the rescue ability of human cDNAs and variants (*Bellen and Yamamoto, 2015*). The odds that this strategy will be effective for the majority of genes seem limited at first glance given the following issues: the test is done with a single cDNA yet two or more protein isoforms are encoded by the vast majority of genes (*Table 1*); there may be issues with expression levels as shown for *UAS-GFP* versus *GFSTF*; timing of protein production may be delayed; and finally, tagging of cDNAs (*HA* or *GFP*) has been documented to impair function for ~20–30% of the tagged cDNAs (*Bischof et al., 2013*; *Sarov et al., 2016*). Nevertheless, as shown in *Table 1*, about 70% of the *UAS-cDNA* constructs were able to rescue lethality, despite the fact that nearly all genes tested encode more than one protein isoform. In addition, no obvious pattern emerged from these data with respect to the presence or absence of a tag (Chi sq. = 0.0004, p=0.98), and no pattern emerged with respect to rescue of lethal mutations, as these genes encode anywhere from 1 to 20 protein isoforms but often could be rescued with a single cDNA (*Table 1*). Establishing that there is complete rescue of all phenotypes, not just lethality, would be time consuming and require detailed studies including longevity, fertility, and numerous behavioral assays beyond the scope of this work. We note that we also previously showed that intronic tagging with *GFSTF* disrupted about 25% of the genes (*Nagarkar-Jaiswal et al., 2015b*). Hence, we recommend that both tagged and untagged cDNAs be tested whenever possible.

In summary, this library provides a set of ~1000 gene-specific *GAL4* drivers for the fly community. We are in the process of creating numerous other *T2A-GAL4* insertions as part of the Gene Disruption Project and we prioritize genes based on the nomination from scientific community through a web site (http://www.flyrnai.org/tools/crimic/web/). The *GAL4/UAS* system is a very well established binary approach and this *T2A-GAL4* library will provide numerous additional tools to survey gene and circuit function in combination with many other existing genetic tools such as *UAS-RNAi*, *UAS-fly* cDNA, *UAS-GCaMP* (*Nakai et al., 2001*), *UAS-ChR* (*Schroll et al., 2006*), *UAS-shi*ts (*Kitamoto, 2001*) and so on. For an estimated 90% of the genes tested, the insertion of *SA-T2A-GAL4-polyA* causes a severe loss-of-function mutation and only three insertions displayed dominant phenotypes out of ~1000 genes tested. Finally, the *T2A-GAL4* flies provide a very useful platform for functional testing of fly as well as human genes and their possible disease variant(s) (*Chao et al., 2017*; *Chen et al., 2016*; *Luo et al., 2017*; *Sandoval et al., 2014*; *Wangler et al., 2017*; *Yoon et al., 2017*).

## Materials and methods

### Fly strains

Fly stocks were maintained on standard cornmeal-yeast-agar medium at 25°C, and on a 12/12 hr light/dark cycle. The MiMIC *and* CRIMIC flies were created in the Bellen lab (see Flypush or *Supplementary file 2*). UAS-2xEGFP, hs-Cre,vas-dφC31, Trojan T2A-GAL4 triplet flies were from Dr. Ben White (*Diao et al., 2015*). The RMCE conversion of MiMICs with *GFSTF* and *T2A-GAL4* cassettes was described in previous studies (*Diao et al., 2015*; *Nagarkar-Jaiswal et al., 2015a*; *Nagarkar-Jaiswal et al., 2015b*). The crossing schemes for CRIMICs are shown in Supplemental Information 1. *btl-GAL4, Ilp2-GAL4, repo-GAL4, gcm-GAL4, UAS-mCD8::GFP, UAS-mCD8::RFP, P [acman]* flies, and *UAS-FLP* flies were obtained from the Bloomington *Drosophila* Stock Center (BDSC, USA). *UAS-if* was from Dr. Celeste Berg (*Beumer et al., 1999*). *NP1243-GAL4, NP2222-GAL4,* and *NP6250-GAL4* are from Kyoto Stock Center (Kyoto DGGR, Japan). Dfs flies were from BDSC or Kyoto DGGR. *UAS-cDNA* flies were from BDSC or FlyORF (Switzerland). *y,w;attP40(y+){nos-Cas9(v+)}/CyO* (*Kondo and Ueda, 2013*) and *y,w;+/+; attP2(y+){nos-Cas9(v+)}* (*Ren et al., 2013*) were isogenized in this work. See *Supplementary file 2* for the genotypes and stock numbers of fly stocks. All references to FlyBase are based on FlyBase 2.0/FB2017_06 (*Gramates et al., 2017*).

### Plasmid construction

*TypeIISRE-attP-FRT-SA-3xStop-SV40-3xP3-GFP-SV40-FRT-attP-TypeIISRE* fragment was synthesized in two parts by GENEWIZ (www.genewiz.com) in the pUC57 vector (pM5 and pM7 were synthesized by GENEWIZ). Next, the ~1.2 kb fragment of *attP-FRT-SA-3xStop-SV40-3xP3* in pM5 was digested with BstXI and EagI. The ~1.3 kb fragment of *GFP-SV40-FRT-attP* in pM7 was digested with EagI and EcoRV. To generate pM14, these two DNA fragments were separated and purified from agarose gel and ligated with pBS-deltaBsaI vector which was digested with BstXI and EcoRV (molar ratio of insert:vector = 5:1). The ligation mix (1 μg/8 μl total DNA + 1 μl 10xT4 DNA Ligase Buffer + 1 μl T4 DNA ligase) was incubated at 16°C overnight then transformed into NEB® Stable *E. coli* competent cells. Cells were raised on ampicillin (50 μg/ml)/LB agar plate at 37°C overnight. pM14 plasmids were checked by double digestion of BstXI and EcoRV.

pM36 was modified from pM14 by removing two *FRT* sites in pM14 by mutagenesis. pM36 was modified from pM14 by sequentially adding 25 nucleotides flanking each of the *attP* sites for sequencing the inserted homology arms and mutating the two *FRT* sites to render them nonfunctional. In brief, a NsiI-EcoRI fragment containing the necessary modifications was cloned by PCR from oligos (DLK256 = taaatATGCATcgatcgtctggtactacattcacgcGTACTGACGGA CACACCGAAGCccc (fwd) and DLK331 = AGAGAGAATTCCTACATGGTAATGT TACTAGAGAA TAGGAACTTCTCGCGCTC (rev)) using pM14 as a template and inserted between the NsiI and EcoRI sites to replace the original pM14 sequence, followed by cloning a XbaI-SphI fragment from pM14 with the necessary modifications for the downstream site using the oligos (DLK332 = TATTC TCTAGAAACATTACCATGTAGTCGCGCTCGCGCGACTGACG (fwd) and DLK255 = GGTAGGAA-GACAACGCGCAGTGAAGGACGAGAGGTAGTACC GCATGCGTACTGACGGACACACCG (rev)) and replacing the pM14 sequences between the XbaI and SphI sites.

pM37 vectors were modified from pM14 by replacing *3xStop* with *T2A-GAL4* of different phases from pT-GEM vectors of the corresponding phase (*Diao et al., 2015*). Briefly the EcoRI-PstI fragment of pM14 was subcloned in pBluescript SK and mutagenized by PCR mutagenesis to replace *3XStop* sequences with AscI restriction enzyme site and subcloned back in pM14 vector. *T2A-GAL4* sequences were PCR amplified from pT-GEM vector and cloned in EcoRI/MfeI and AscI sites in mutated pM14, generating pM14 *T2A-GAL4* vector. AscI-SbfI fragment of *T2A-GAL4* was resynthesized to remove Type IIS RE sites by substituting base pairs corresponding to Type IIS REs with synonymous mutations eliminating the sites. The resulting fragment was subcloned in pM14 *T2A-GAL4* vector. pM14 and pM36 vectors were found to be unstable in bacteria, frequently recombining out the *3XP3-GFP* cassette. Further analysis showed that *3XP3* promoter fragment of pM14 and pM36 was 290 bps longer than other vectors that use the same marker. Shortening this fragment by PCR and replacing the AscI-FseI fragment with the shortened fragment improved stability of the vector in bacteria, creating the pM37 vector. Sequences of pM14, pM36 and pM37 can be found in *Supplementary file 3*.

## CRIMIC production

We analyzed the introns of all protein-coding genes of *Drosophila melanogaster* annotated at Fly-Base and selected the genes that have at least one CDS intron that is >100 bp and is shared by all isoforms. Based on FlyBase release 6.16, there are 5822 protein-coding genes that meet these criteria. Then, we removed the genes that are covered by the MIMIC Gold collection and prioritized the genes if their human ortholog(s) are disease-related (*Hu et al., 2011*). We also prioritized genes based on the nomination from scientific community through a web site (http://www.flyrnai.org/tools/crimic/web/). sgRNA targeting the qualified CDS introns were selected based on their efficiency score and specificity annotated at Find CRISPR Tool (*Ren et al., 2013*). The homology arms upstream or downstream of the cutting site were designed using Primer3 (*Untergasser et al., 2012*). We required that the homology arms are between 500 and 1200 bp in length, less than 40 bp apart from each other, and free of one or more of the three restriction enzymes (BsaI, BbsI, BsmBI) used for cloning.

Donor constructs were generated as previously described (*Housden and Perrimon, 2016*). Briefly, homology arms were PCR amplified from genomic DNA using Q5 or Phusion polymerase (NEB), run on an agarose gel and purified with the QIAquick Gel Extraction Kit (Qiagen). The homology arms, pBH donor vector and pM14/pM36/pM37 cassette were combined by Golden Gate assembly (*Engler et al., 2008*) using the appropriate type IIS restriction enzyme (BbsI, BsaI, or BsmBI). The resulting reaction products were transformed into Stbl3 or TOP10 Chemically Competent Cells (ThermoFisher), and plated overnight under kanamycin selection. Colonies were cultured for 24 hr at 30°C and DNA prepared by miniprep. The entire homology arm sequence and 300–500 bps of the adjacent cassette sequence were verified prior to injection.

sgRNA constucts were generated as previously described (*Housden et al., 2016*). Briefly, sense and antisense oligos containing the 20 bp guide target sequence were annealed and phosphorylated with T4 Polynucleotide Kinase (NEB), then inserted between BbsI sites in the pl100 sgRNA expression vector (*Ren et al., 2013*). Ligation products were transformed into TOP10 Competent Cells (ThermoFisher), and plated overnight. Colonies were cultured, DNA prepared by miniprep, and sequences verified prior to injection. We injected a mix of 25 ng/µl sgRNA and 150 ng/µl donor DNA in isogenized fly embryos of the following genotypes *y,w; attP40(y+){nos-Cas9(v+)}/CyO* (*Kondo and Ueda, 2013*) and *y,w; +/+; attP2(y+){nos-Cas9(v+)}* (*Ren et al., 2013*) to generate CRIMIC insertions (*Housden et al., 2016*; *Housden and Perrimon, 2016*).

## PCR validation

For validation of MiMIC conversion and CRIMIC cassette insertion events, the genomic DNA was extracted from ~20 adult flies using the PureLink Genomic DNA Mini Kit (Invitrogen). For MiMIC conversions, four reactions of PCR were performed with tag-specific primers and MiMIC specific primers as described previously (*Diao et al., 2015*; *Venken et al., 2011a*). The PCR reaction mix was: 1 µl genomic DNA (~10 ng), 1 µl primer 1 (10 µM), 1 µl primer 2 (10 µM), 4.5 µl H2O, and 7.5 µl GoTaq Green Master Mix (Promega). Hot start PCR conditions in C100 Touch Thermal Cycler (Bio-Rad) were: denaturation at 95° for 1 min, 34 cycles at 95° for 30 s, 56° for 30 s and 72° for 60 s, and post-amplification extension at 72° for 10 min. For CRIMIC cassette insertion, two reactions of PCR were performed with target-specific primers (see our website at Flypush) and attP-R primer (5'-CCCCAGTTGGGGC-3') (*Figure 2—figure supplement 1*). PCR reaction mix was: 1 µl genomic DNA (~10 ng), 1 µl primer 1 (10 µM), 1 µl primer 2 (10 µM), 4.5 µl $H_2O$, and 7.5 µl GoTaq Green Master Mix (Promega). Hot start PCR conditions in C100 Touch Thermal Cycler (Bio-Rad) were: denaturation at 95° for 1 min, 40 cycles at 95° for 30 s, 56° for 30 s and 72° for 2 min 30 s, and post-amplification extension at 72° for 10 min.

## pM37 cassette excision

Virgin female pM37 flies were collected and crossed with male flies carrying a *UAS-FLP* on the third chromosome. The adult eyes of *pM37/+;+/+;UAS-FLP/+* for insertions in *Dsor1*, *Raf* and *Marf* were imaged with a fluorescent microscope (Zeiss SteREO Discovery.V20).

## Confocal imaging

Confocal imaging was performed as described previously (*Lee et al., 2011*). In brief, dissected adult brains or VNCs were fixed in 4% paraformaldehyde/1xPBS at 4°C overnight, transferred to 2% Triton X-100/1xPBS at room temperature, vacuumed for 1 hr and left overnight in the same solution at 4°C. The larvae brains or other tissues were fixed in 4% paraformaldehyde/1 xPBS at 4°C for at least 2 hr, transferred to 0.5% Triton X-100/1xPBS at 4°C for overnight. For immunostaining, the samples were blocked in 10% NGS/0.5% Triton X-100/1xPBS and incubated with primary antibodies (1:50 ~ 200 dilution) at 4°C for overnight with shaking, then washed with 0.5% Triton X-100/1xPBS for 5 min three times. The secondary antibodies conjugated with Alexa-488 or Alexa-647 (Jackson ImmunoResearch) were diluted 1:100 ~ 500 in 0.5% Triton X-100/1xPBS and incubated with samples at 4°C for overnight with shaking. For immunostaining of GFP, the samples were incubated with anti-GFP antibody conjugated with FITC (1:500) (Abcam) in 1xPBS with 0.5% Triton X-100 for overnight. Samples were cleared and mounted in RapiClear (SunJin Lab Co.) and imaged with a Zeiss LSM 880 Confocal Microscope under a 20x or 40x C-Apochromat water immersion objective lens.

## Acknowledgements

We thank Ben White, Celeste Berg, the Bloomington *Drosophila* Stock Center, FlyORF, and the Kyoto Stock Center for fly stocks, and the Developmental Studies Hybridoma Bank for antibodies. We thank Travis Johnson for reading the manuscript and giving us very helpful suggestions. We thank Jiangxing Lv for brain dissections, imaging, and PCR, and Qiaohong Gao, Zhihua Wang, Junyan Fang, Liwen Ma and Lily Wang for generating and maintaining MiMIC/CRIMIC *T2A-GAL4* fly stocks. Confocal microscopy was performed in the BCM IDDRC Neurovisualization Core, supported by the NICHD (U54HD083092). This research was supported by NIH grants R01GM067858 and R24OD022005. HJB receives support from the Robert A and Renee E Belfer Family Foundation and the Huffington Foundation. SY is supported by the Alzheimer's Association (NIRH-15–364099), Simons Foundation (SFARI- 368479), Naman Family Fund, Caroline Wiess Law Fund, and NIH (U54NS093793). JZ, YHu, SEM and NP receive support from NIGMS (GM067761 and GM084947) and SEM from the Dana Farber/Harvard Cancer Center (NIH 5 P30 CA06516). NP, ACS and HJB are investigators of the Howard Hughes Medical Institute.

## Additional information

### Competing interests

Hugo J Bellen: Reviewing editor, *eLife*. Allan C Spradling: Reviewing editor, *eLife*. The other authors declare that no competing interests exist.

### Funding

| Funder | Grant reference number | Author |
|---|---|---|
| National Institutes of Health | R01GM067858 | Pei-Tseng Lee |
| National Institute of General Medical Sciences | GM067761 | Jonathan Zirin<br>Yanhui Hu<br>Stephanie E Mohr |
| Howard Hughes Medical Institute | | Karen L Schulze<br>Yuchun He<br>Hongling Pan<br>Stephanie E Mohr<br>Robert W Levis<br>Allan C Spradling<br>Norbert Perrimon<br>Hugo J Bellen |
| Dana-Farber/Harvard Cancer Center | 5 P30 CA06516 | Stephanie E Mohr |
| Huffington Foundation | | Shinya Yamamoto |
| Alzheimer's Association | NIRH-15-364099 | Shinya Yamamoto |

| Simons Foundation | 368479 | Shinya Yamamoto |
| Naman Family Fund for Basic Research | | Shinya Yamamoto |
| Caroline Wiess Law Fund | | Shinya Yamamoto |
| National Institutes of Health | U54NS093793 | Shinya Yamamoto |
| National Institute of General Medical Sciences | GM084947 | Norbert Perrimon |
| Eunice Kennedy Shriver National Institute of Child Health and Human Development | U54HD083092 | Hugo J Bellen |
| Robert A. and Renee E. Belfer Family Foundation | | Hugo J Bellen |
| National Institutes of Health | R24OD02205 | Hugo J Bellen |

The funders had no role in study design, data collection and interpretation, or the decision to submit the work for publication.

## Author contributions

Pei-Tseng Lee, Conceptualization, Data curation, Formal analysis, Writing—original draft, Project administration, Writing—review and editing; Jonathan Zirin, Karen L Schulze, Robert W Levis, Data curation, Project administration, Writing—review and editing; Oguz Kanca, David Li-Kroeger, Conceptualization, Data curation, Project administration, Writing—review and editing; Wen-Wen Lin, Stephanie E Mohr, Shinya Yamamoto, Project administration, Writing—review and editing; Rong Tao, Colby Devereaux, Yanhui Hu, Verena Chung, Ying Fang, Yuchun He, Hongling Pan, Ming Ge, Zhongyuan Zuo, Benjamin E Housden, Project administration; Allan C Spradling, Writing—review and editing; Norbert Perrimon, Supervision, Funding acquisition, Project administration, Writing—review and editing; Hugo J Bellen, Conceptualization, Data curation, Supervision, Funding acquisition, Writing—original draft, Project administration, Writing—review and editing

## Author ORCIDs

Pei-Tseng Lee (iD) http://orcid.org/0000-0002-7501-7881
Karen L Schulze (iD) http://orcid.org/0000-0002-1368-729X
Benjamin E Housden (iD) http://orcid.org/0000-0001-9134-4279
Stephanie E Mohr (iD) http://orcid.org/0000-0001-9639-7708
Shinya Yamamoto (iD) http://orcid.org/0000-0003-2172-8036
Norbert Perrimon (iD) http://orcid.org/0000-0001-7542-472X
Hugo J Bellen (iD) http://orcid.org/0000-0001-5992-5989

## Decision letter and Author response

Decision letter https://doi.org/10.7554/eLife.35574.022
Author response https://doi.org/10.7554/eLife.35574.023

## Additional files

### Supplementary files

• Supplementary file 1. CRISPR crosses.
DOI: https://doi.org/10.7554/eLife.35574.017

• Supplementary file 2. Information of MiMIC/CRIMIC lines, fly stocks, complementation results and resource.
DOI: https://doi.org/10.7554/eLife.35574.018

• Supplementary file 3. DNA sequences of CRIMIC donor vectors.
DOI: https://doi.org/10.7554/eLife.35574.019

• Transparent reporting form
DOI: https://doi.org/10.7554/eLife.35574.020

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
