## [Decision Letter]

Thank you for submitting your article "A gene-specific *T2A-GAL4* library for *Drosophila*" for consideration by *eLife*. Your article has been reviewed by two peer reviewers, and the evaluation has been overseen by K VijayRaghavan as the Senior and Reviewing Editor. The reviewers have opted to remain anonymous.

The reviewers have discussed the reviews with one another and the Reviewing Editor has drafted this decision to help you prepare a revised submission before acceptance. Given the nature of this contribution, we believe it would be more appropriate considered as a Tools and Resources paper not a Research article. Let us know if you have any concerns about reclassifying the paper in the TR category.

Summary:

The manuscript by Lee et al. describes the latest iteration of tools and transgenic fly lines generated by the Spradling, Perrimon, and Bellen labs in conjunction with the *Drosophila* Gene Disruption Project (GDP), which has as its goal the introduction of mutations into most or all genes in the fruit fly. In this manuscript, the authors build on strategies that in addition to being mutagenic, also permit analysis of gene and protein expression at the cellular and sub-cellular levels. The emerging GDP toolkit described thus allows the increasingly comprehensive characterisation of gene function and expression in *Drosophila*. Importantly, the new fly lines and tools introduced in this manuscript specifically permit the functional characterisation of cells that express a gene of interest (GOI) because of the insertion of a *T2A-GAL4* cassette into the coding intron of the GOI. As has been previously shown by other groups, coupling *Gal4* expression to a GOI allows the expression of transgenes in the same cells that express the GOI to effect various manipulations. The many new transgenic lines introduced in the manuscript constitute an important new resource for the *Drosophila* research community and the pM37 CRIMIC construct described here will likewise be useful in the generation of new gene- and cell-type specific fly lines, not only by the GDP, but by individual researchers. The authors do a nice job characterising both their methodology and the new fly lines, providing abundant examples of their utility for a number of applications.

The paper presents a valuable resource of strains and a well-constructed system (CRIMIC) to analyze gene functions (in these specific strains and beyond). The work is carefully done and properly documented. The experimental possibilities of the presented system are demonstrated one after another, thus exemplifying its capabilities and limitations. These resources (MIMIC and CRIMIC strains and cassettes) provide additional tools to more carefully study gene functions, independent of whether a MIMIC line is currently available or not. This CRISPR-mediated approach principally allows to target many more genes, provided there is an appropriate entry site available (i.e. intron). The Discussion is written in a very balanced way, not overhyping the work, simply reviewing pros and cons, very helpful.

However, there are some important concerns, most which can in the main be addressed by careful writing and referencing. These should be speedily addressed for the manuscript to be ready for acceptance.

Essential revisions:

1) A major virtue of this manuscript is its scale. Over 1,000 new transgenic lines are described, many of which are already publicly available through the Bloomington *Drosophila* Stock Center. One point of potential confusion is that the lines deposited at Bloomington are designated as "*Trojan GAL4*" lines-following the terminology of Diao et al. (2015) who introduced the technology used to generate them – but the manuscript doesn't explicitly refer to them by this name. Doing so would be useful, as the inconsistency may cause confusion to readers and potential end-users looking for the lines.

2) Although the *T2A-GAL4* technology is applied with unprecedented scope in this manuscript, its application, efficacy, and some of its limitations have been already relatively well-established in the literature-a fact that is too understated in the manuscript. For example, the authors state that "few genes have been reported to be tagged with a *T2A-GAL4* cassette" and cite 5 papers, but a quick literature search for citations to the papers that introduced the technology (Diao et al., 2015 and Gnerer et al., 2015) reveals a total of 11 papers and at least 34 genes with *T2A-GAL4*s inserted. In numerous cases, the fidelity of expression of the *T2A-GAL4* line is confirmed, the mutagenic effects of insertion are shown, and examples of rescue by UAS-transgenes are reported. In these respects, this manuscript simply confirms on a larger scale what has already been shown and it makes little sense to treat the effects of *T2A-GAL4* insertion as matters of speculation remaining to be demonstrated, as occurs in several places (e.g. *T2A-GAL4* insertion "should allow determination of expression patterns", "should cause a severe loss-of-function mutation", "should" or "may allow rescue" of phenotypes when driving UAS-cDNAs.) These statements should be re-phrased in a way that takes into account previous observations in the literature, which should also be more fully referenced.

3) A previously noted problem with some *T2A-GAL4* conversions of MiMIC insertions is that they fail to result in viable heterozygous offspring. It would be interesting to know in these experiments what the success/failure rate in generating viable stocks containing the conversions was.

4) The individual (lettered) panels of Figures 1, 2, and 3 are too crowded, and should be separated from each other more so as not to run together.

5) The manuscript does not discuss the T-GEM construct of Diao et al. (2015) which was previously introduced to insert *T2A-GAL4* into coding introns using the CRISPR/Cas system. It would be beneficial for readers if in the first paragraph of the subsection “Coding intronic insertions of the *SA-T2A-GAL4-polyA* cassette generate loss-of-function mutations for ~90% of insertions” discussing the design of the pM37 CRIMIC, the authors briefly compared this construct to the T-GEM construct – particularly to make readers aware of the fact that the CRIMIC construct has the potential benefit of being revertible by Flp. Also, it might be useful to readers to cite in the context of the fifth point (i.e. the ability to insert any DNA flanked by attB sites into a CRIMIC) some of the papers which describe attB-flanked constructs, such as the *T2A-Gal80* construct of Gnerer et al. (2015) and the *T2A-LexA, QF2*, and *Split-Gal4* components of Diao et al. (2015).

---

## [Author Response]

Essential revisions:1) A major virtue of this manuscript is its scale. Over 1,000 new transgenic lines are described, many of which are already publicly available through the Bloomington Drosophila Stock Center. One point of potential confusion is that the lines deposited at Bloomington are designated as "Trojan GAL4" lines-following the terminology of Diao et al. (2015) who introduced the technology used to generate them – but the manuscript doesn't explicitly refer to them by this name. Doing so would be useful, as the inconsistency may cause confusion to readers and potential end-users looking for the lines.

We added the "*Trojan GAL4*" in main text and title of a subsection (Introduction, third paragraph and subsection “Comparison of *GFSTF* and *Trojan-GAL4* tagging of MiMIC-containing genes”) to emphasize that we use *Trojan GAL4* cassettes generated by Diao et al. (2015) to convert 619 MiMICs to *T2A-GAL4s* in this manuscript.

2) Although the T2A-GAL4 technology is applied with unprecedented scope in this manuscript, its application, efficacy, and some of its limitations have been already relatively well-established in the literature-a fact that is too understated in the manuscript. For example, the authors state that "few genes have been reported to be tagged with a T2A-GAL4 cassette" and cite 5 papers, but a quick literature search for citations to the papers that introduced the technology (Diao et al., 2015 and Gnerer et al., 2015) reveals a total of 11 papers and at least 34 genes with T2A-GAL4s inserted. In numerous cases, the fidelity of expression of the T2A-GAL4 line is confirmed, the mutagenic effects of insertion are shown, and examples of rescue by UAS-transgenes are reported. In these respects, this manuscript simply confirms on a larger scale what has already been shown and it makes little sense to treat the effects of T2A-GAL4 insertion as matters of speculation remaining to be demonstrated, as occurs in several places (e.g. T2A-GAL4 insertion "should allow determination of expression patterns", "should cause a severe loss-of-function mutation", "should" or "may allow rescue" of phenotypes when driving UAS-cDNAs.) These statements should be re-phrased in a way that takes into account previous observations in the literature, which should also be more fully referenced.

We re-wrote "few genes have been reported to be tagged with a *T2A-GAL4* cassette" to "about 50 genes have been reported to be tagged with a *Trojan-GAL4* cassette" and added some references. We also changed the “should” to “could” in the third paragraph of the Introduction. However, we were unable to find data on several other issues mentioned by the reviewer. Indeed, very few *T2-GAL4* insertions so far have been rescued with the UAS-cDNA of the corresponding gene and those that have been rescued were done in our lab. Hence, we kept out original word usage. Also, very few *T2A-GAL4* insertions have been studied with respect to their impact on gene function. So we kept our original word usage.

3) A previously noted problem with some T2A-GAL4 conversions of MiMIC insertions is that they fail to result in viable heterozygous offspring. It would be interesting to know in these experiments what the success/failure rate in generating viable stocks containing the conversions was.

The reviewer makes a good point and we have now added the following data. We now state that the *MiMIC T2A-GAL4* conversion rate is 68.1% (543/797) upon a single attempt. The rate upon second attempts is 41.1% (76/185). Hence, out of 797 attempted genes, 109 failed twice. These data are now included in the first paragraph of the Discussion.

4) The individual (lettered) panels of Figures 1, 2, and 3 are too crowded, and should be separated from each other more so as not to run together.

We increased the space between individual panels for the Figure 1 and 2. We keep the Figure 3 since it looks fine in our opinion.

5) The manuscript does not discuss the T-GEM construct of Diao et al. (2015) which was previously introduced to insert T2A-Gal4 into coding introns using the CRISPR/Cas system. It would be beneficial for readers if in the first paragraph of the subsection “Coding intronic insertions of the SA-T2A-GAL4-polyA cassette generate loss-of-function mutations for ~90% of insertions” discussing the design of the pM37 CRIMIC, the authors briefly compared this construct to the T-GEM construct – particularly to make readers aware of the fact that the CRIMIC construct has the potential benefit of being revertible by Flp. Also, it might be useful to readers to cite in the context of the fifth point (i.e. the ability to insert any DNA flanked by attB sites into a CRIMIC) some of the papers which describe attB-flanked constructs, such as the T2A-Gal80 construct of Gnerer et al. (2015) and the T2A-LexA, QF2, and Split-Gal4 components of Diao et al. (2015).

We now mention the T-GEM explicitly and added a sentence ….“similar to the T-GEM vector developed by Diao et al. (Diao et al., 2015). However, we added flanking FRT sites to allow excisions of the cassette with Flippase.”